# Exogenous Phytohormones and Fertilizers Enhance *Jatropha curcas* L. Growth through the Regulation of Physiological, Morphological, and Biochemical Parameters

**DOI:** 10.3390/plants11243584

**Published:** 2022-12-19

**Authors:** Rahmatullah Jan, Murtaza Khan, Muhammad Adnan, Sajjad Asaf, Saleem Asif, Kyung-Min Kim, Waheed Murad

**Affiliations:** 1Department of Applied Biosciences, Graduate School, Kyungpook National University, Daegu 41566, Republic of Korea; 2Coastal Agriculture Research Institute, Kyungpook National University, Daegu 41566, Republic of Korea; 3Department of Horticulture and Life Science, Yeungnam University, Gyeongsan 38541, Republic of Korea; 4Department of Botanical and Environmental Sciences, Kohat University of Science and Technology, Kohat 26000, Pakistan; 5Natural and Medical Sciences Research Center, University of Nizwa, Nizwa 616, Oman; 6Department of Botany, Abdul Wali Khan University, Mardan 23200, Pakistan

**Keywords:** biodiesel, gibberellin, growth parameters, indole acetic acid, *Jatropha curcas*, proximate composition

## Abstract

*Jatropha curcas* L. is a perennial plant, that emerged as a biodiesel crop attracting the great interest of researchers. However, it is considered a semi-wild plant and needed to apply crop-improving practices to enhance its full yield potential. This study was conducted to improve the growth and development of the *J. curcas* plant by exogenous application of Gibberellic acid (GA), indole acetic acid (IAA), and fertilizer (nitrogen, phosphorus, potassium (NPK)). The experiment was conducted in pots in triplicate and 100 ppm and 250 ppm of GA and IAA were applied separately while NPK was applied in two levels (30 and 60 g/pot). The results revealed a significant difference in growth parameters with the application of hormones and fertilizer. The highest shoot length (47%), root length (63%), root fresh weight (72%), and root dry weight (172%) were shown by plants treated with GA 250 ppm. While plants treated with NPK 60 g showed the highest increases in shoot fresh weight and shoot dry weight compared to control plants. The highest increase in leaves number (274%) and branches number (266%) were shown by the plants treated with GA 100 ppm and GA 250 ppm, respectively, while GA 250 ppm and IAA 250 ppm highly increased stem diameter (123%) and stem diameter was also shown by GA 250 ppm-treated plants. NPK 60 g highly increased proximate composition (protein content, carbohydrate, fat, moisture content, and ash content) compare with hormones and control plants. Our results concluded the optimized concentration of IAA, GA, and NPK significantly increases *J. curcas* growth vigor.

## 1. Introduction

*Jatropha curcas* L. is a perennial shrub, belonging to the family Euphorbiaceae. It is about 5 m tall and has smooth grey bark, leaves are large and usually pale green. Fruits are produced in winter or throughout the year depending on temperature and soil humidity. The indications show that it originated from South and Central America and some other parts of the tropical and subtropical regions of Africa and Asia [1]. Specifically, it is grown in Benin, Brazil, China, Egypt, Ethiopia, Ghana, Guinea, India, Madagascar, Mali, Mexico, Mozambique, Namibia, Senegal, South Africa, Sudan, Tanzania, Uganda, Zambia, and Zimbabwe [2]. Its name indicates that it is used as a medicinal plant in the Portuguese in the 16th century, as its name is derived from the Greek word “iatros” which means doctor, and “trophe” which means food [3]. Literature shows that the Portuguese established the plantation of *J. curcas* for the first time to make soap, lamp oil, and other medicine [1]. 

However, *J. curcas* is exotic to Pakistan and introduced in 2007 to Karachi. It is an oilseed plant that does not compete with food crops and its seed contains 34–60% oil contents [3,4]. It has the ability to adapt to a high range of agro-climatic conditions. It grows in gravelly, sandy, degraded, acidic, and poor stony soils. It is easy to establish, grows quickly, drought tolerant, grow well in low rainfall condition, and can be used to reclaim eroded areas [5]. Therefore, it is suggested for cultivation on poor degraded soil. However, it shows stunt growth under heavy metals stress [6]. It can survive in long dry periods and it is considered to be well adapted to arid and semi-arid conditions. In the initial stage, it requires a large amount of water but after maturity, it may survive without water for sixty days. In the rainfed area, it required 250 up to 3000 mm/year of rainfall for best growth [7]. Gadallah and Sayed (2001) reported that the exogenous application of hormones increases *J. curcas* resistance against environmental constraints and promotes plant growth and development, however, the effectiveness of growth regulators depends on their concentration and the method of application [8]. 

To promote plant growth and development in non-suitable conditions, phytohormones or macronutrients are important candidates. Trail studies indicated that phytohormones and fertilizers increase plant growth and development under stress conditions. Gibberellin (GA) increases plant growth, leaves number, bud formation, cell division and elongation, and flowering [9,10]. Indol acetic acid (IAA) is also involved in several developmental processes such as the development of vascular tissue, cell elongation, and apical dominance [11]. The growth regulator enhances the growth of *J. curcas* by regulating its morpho-physiological and biochemical processes [8]. Besides hormones, many other parameters also affect the growth of *J. curcas*. Fertilizer is one of the essential parameters which promote plant development in a harsh environment. Nitrogen, phosphorous, and potassium (NPK) are the main nutrients needed for plant growth and development. Without providing adequate NPK, plants cannot reach maximum growth [12]. The NPK supplementation increases the biomass and oil contents in *J. curcas,* which shows the importance of fertilizers for the growth and development of *J. curcas*. Being a biofuel plant, *J. curcas* attracted the researcher’s attention. As it grows well in drought and high temperatures; therefore, it is suitable for arid regions where it is not competing with the crops for water and land. 

Due to the potential demand and better opportunities, cultivation of *J. curcas* appears viable. It adapts well to marginal lands and the large-scale cultivation on wasteland with low water and rainfall could generate employment and increase the income of the locality. However, the ability of *J. curcas* to grow in marginal land and dry soil has not been properly explored [13]. Several studies have been conducted to evaluate the *J. curcas* performance under low water availability in marginal land [13]. In order to understand the plant growth performance in the marginal land of Kohat, Khyber Pakhtun Khwa, Pakistan, region. Kohat is an arid region containing a large quantity of marginal land, which is suitable for *J. curcas* cultivation. Based on *J. curcas* plant adaptation to marginal land, we conducted a trail base studyin the Kohat region (arid region) district in Khyber Pakhtunkhwa, Pakistan. However, the result of plant growth was not satisfactory under normal growth conditions (data not published). To further investigate the root cause of our previous project failure, we performed the current study to investigate the possible way to promote the growth of *J. curcas* in the same region. In the current study, we focus on exogenous hormonal application and fertilizer supplementation. We hypothesized that GA, IAA and NPK promote *J. curcas* growth and development by regulation of morphological, physiochemical and biochemical characteristics of *J. curcas*. Therefore, the current study aimed to investigate that the exogenous hormones and fertilizer induces morphological, physiological, and biochemical parameters of the *J. curcas* plant and enable it to grow and develop in arid regions. Based on our study, we can further suggest the cultivation of the *J. curcas* plant in the barren land after phytohormones and fertilizers supplementation. This study is of great importance and thereby suggests that *J. curcas* growth and development increases with the exogenous application of phytohormones and fertilizers. *J. curcas* is an economically important plant and unlike crop plants, it does not need significant attention and does not compete with other crops for land and water. The current study added a new insight to the field that application of phytohormones and fertilizers can promote *J. curcas* growth and development in the marginal land under limited water availability. 

## 2. Results 

### 2.1. Phytohormones and Fertilizer Enhances J. curcas Growth Vigor 

In the present study, it is investigated that IAA, GA, and NPK increased *J. curcas* shoot length (Figure 1). IAA 100 ppm and IAA 250 ppm increased significantly the shoot length up to 25% and 29%, respectively, as compared to control plants after four months of growth. The GA 100 ppm, GA 250 ppm, and NPK 30 g significantly increased the shoot length up to 17, 47, and 41%, respectively, after four months of growth, compared with control plants (Figure 1A). The higher concentration of IAA, GA, and NPK also increased shoot fresh and dry weight (Figure 1B,C). The results indicated that IAA 250 ppm significantly increased fresh and dry weight up to 141 and 183%, respectively. GA 250 ppm increased shoot fresh and dry weight up to 141 and 179%, respectively, while NPK 60 g increased by 185 and 267%, respectively, compared to control plants. NPK-applied plants showed higher fresh and dry weight than the GA and IAA. On the other hand, hormones and fertilizer also induced root biomass. The IAA 250 ppm, GA 100 ppm, GA 250 ppm, and NPK 60 g significantly increased root length up to 31, 50, 63, and 40%, respectively (Figure 1D). Root fresh weight was significantly increased by 21% by IAA 250 ppm, 72% by GA 250 ppm, and 30% by NPK 60 g compared with control plants (Figure 1E). The root dry weight followed the same pattern of the root fresh weight. The IAA 250 ppm increased the root dry weight to 81%, GA 250 ppm increased to 172%, and NPK 30 g and 60 g increased to 63 and 118%, respectively (Figure 1F). We investigated that, among the hormones and fertilizer, the highest shoot length was shown by the GA 250 ppm-treated plants, while the highest shoot fresh and dry weight was shown by NPK 60 g-treated plants. However, the GA 250 ppm-treated plant showed the highest increase in root length, root fresh, and dry weight which shows that GA can induce *J. curcas* root growth and development. 

### 2.2. Phytohormones and Fertilizer Treatment Promotes Leaf and Branch Number and Stem Diameter

In the current study, we found that IAA, GA, and NPK efficiently stimulated *J. curcas* number of leaves and branches, and stem diameter in four months (Figure 2). Both low and high concentrations of hormones and NPK application significantly increased the number of leaves, branches, and stem diameter as compared to control plants. IAA 100 ppm and 250 ppm increased leaves number by 262 and 211%, respectively, while branches number increased by 200 and 160%, respectively, and stem diameter increased by 100 and 123%, respectively. GA 100 ppm and 250 ppm increased leaves number up to 274 and 211%, branches number 229 and 260%, and stem diameter 109 and 123%, respectively, as compared to control plants after four months. The 30 g and 60 g of NPK application increased leaves number by 199 and 211%, branches number by 129 and 229%, and stem diameter by 76 and 80%, respectively, after four months of treatment. Among the IAA, GA, and NPK, IAA 100 ppm and GA 250 ppm showed the highest number of leaves increase while GA 250 and NPK 60 g showed the highest induction of branches. Whereas the highest increase in stem diameter was found in IAA 250 ppm and GA 250 ppm treated plants followed by GA 100 ppm. However, the lowest stem diameter and branches number were found in the NPK 30 g treated plants, whereas the lowest leaves number were found in the IAA 100 ppm, GA 250 ppm and NPK 60 g treated plants (equally reduced 211%). These results show that the optimized concentration of IAA, GA, and NPK is needed to increase *J. curcas* leaves, branches, and stem diameter. 

### 2.3. Phytohormones and Fertilizers Promote J. curcas Plant Growth by Regulating Proximate Compositions 

In the current study, we determined the protein, carbohydrate, and fat contents after four months of treatment of IAA, GA, and NPK in different concentrations. The higher concentration of hormones and NPK such as IAA 250 ppm, GA 250 ppm, and NPK 60 g showed a significant increase of protein, carbohydrates, and fat contents compared with control plants (Figure 3). The NPK 60 g treated plants showed the highest increase (34%) in protein contents followed by IAA 250 ppm (27%), and GA 250 ppm (25%) compared with control plants (Figure 3A). IAA 100 ppm also increased significantly the protein content up to 20%, which indicates that increasing the IAA concentration can increase protein contents. Carbohydrates were significantly increased by 61% by NPK 60 g followed by GA 250 ppm 42% and IAA 250 ppm 30%, compared with control plants (Figure 3B). Similarly, the highest increase (46%) in fat contents was found in NPK 60 g treated plants followed by IAA 250 ppm (39%) and GA 250 ppm (32%), compared with control plants (Figure 3C). The highest moisture and ash contents were found in NPK 60 g treated plants followed by IAA 250 ppm and GA 250 ppm treated plants compared with control plants (Figure 3E,F). These results suggested that phytohormones and NPK treatment enhance the growth of *J. curcas* by induction of proximate compositions. Among the phytohormones and NPK, it was found that NPK application highly increased the overall proximate composition compared with IAA and GA. These results show that fertilizers can better increase the *J. curcas* succession in temperate and barren land compared with hormones. 

## 3. Discussion 

In the present study, we reported that IAA, GA, nitrogen, phosphorus, and potassium (NPK) act as growth enhancers of the *J. curcas* plant. We provided physiological, morphological, and biochemical evidence that *J. curcas* growth and development were enhanced by the application of hormones and NPK. Generally, the *J. curcas* plant grows in almost all kinds of soil and drought conditions; however, hormones and fertilizers increase their growth and development. The applied hormones (GA, IAA) and NPK greatly influenced the plant’s morphological, physiological, and biochemical traits. 

IAA is an essential auxin with significant in vivo roles such as stem growth, root growth, stem cambium cell activation, and lateral bud formation [14,15]. In most plants, IAA is the supreme active form of auxin. Recent studies revealed that IAA application has positive effects on growth parameters such as plant root and shoot length, fresh and dry weight, number of leaves, chlorophyll contents, carbohydrates, amino acids, and phenolic contents [16,17]. Our results also showed that IAA increased plant growth parameters (shoot length, root length, fresh and dry weight, branches and leaves number, and stem diameter) in *J. curcas* plants compared with control plants. In a recent study, it is predicted that phytohormones including IAA alter the sugar metabolism which is responsible for the modulation of biological processes that are involved in plant growth promotion [18,19]. IAA interacts with sucrose and alters plant morphogenesis which regulates leaf morphology [20]. In addition, IAA increases endogenous GA accumulation, which is a prominent growth regulator [18]. In our study, we found that IAA increased leaves and branches number. The possible reason for the increased number of leaves and branches number might be the suppression of the ABA hormone. ABA generally inhibits growth or keeps the apical tissue dormant while IAA reduces ABA and increases GA which breaks dormancy and results in plant growth and increasing branches number [18]. Besides other growth parameters, we also found that the IAA-treated plant’s stem diameter was increased significantly than the control plants, which predicts that IAA is efficiently involved in vascular cell division and differentiation. It is reported that IAA increases *Persea americana* cell differentiation which increases vascular vessel density [21]. In *Glycin max*, IAA stimulated plant height, leaves number and area, number of branches, and seed per plant [22]. However, different plants show different reactions to IAA in various concentrations. In the *J. curcas* plant, between the IAA 100 ppm and 250 ppm, the IAA 250 ppm showed a higher increase in morphological, physiological, and biochemical parameters. However, IAA 100 ppm increased the *G. max* height more than that of 200 pmm [22]. Foliar application of IAA on cowpea plants also increased plant height, fresh and dry weight, number of branches, number of leaves, and yield components [23]. As far as IAA is concerned to plant height, some researchers provided evidence that IAA promoted GA synthesis. For instance, it is reported that exogenous IAA increased GA1 and GA_3_ biosynthesis through the activation of GA_1_ and GA_3_ synthesis enzymes [24,25]. IAA not only induced morphological and physiological parameters of the *J. curcas* plant, but also induced their proximate composition (protein, carbohydrate, fat, moisture, and ash contents) compared to control plants. To the best of our knowledge, there is no data published on the effect of IAA and GA on the proximate composition of *J. curcas*; however, researchers investigated the effects of IAA and GA on the proximate composition of various plants. A recent study investigated that, exogenous application of IAA enhanced the photosynthesis rate in *Gossypium hirsutum* which resulted in increased biomass including fresh and dry weight [18]. In the *Balanites aegyptiaca* plant, different concentrations of IAA increased total protein and carbohydrates when compared to non-treated plants [26]. IAA also increased fresh and dry weight, relative water contents (moisture contents), and chlorophyll contents in white clover plants [24]. These results are in line with our findings, which show that IAA is involved in the biosynthesis of protein and carbohydrates and increases moisture contents in *J. curcas*.

GA is a key hormone that induces many parameters of plant growth and development such as seed germination, shoot elongation, leaf expansion, and flower and fruit development [27]. In our study, we found that GA exogenous application increased shoot and root length, fresh and dry weight, and proximate composition percentage in *J. curcas* plants compared with control plants. Plant biomass accumulation is associated with endogenous GA accumulation. GA enhances carbonic anhydrase activity which promotes CO_2_ fixation in photosynthesis as this enzyme takes part in the hydration of CO_2_ and is strictly related to chloroplast [28]. This phenomenon provides enough CO_2_ to the site of fixation and increases the photosynthesis rate which as a result increases biomass accumulation [29,30]. In comparison to IAA and NPK, GA greatly influenced the shoot length, root length, and root fresh and dry weight (Figure 1). Researchers evaluated the basic mechanism of GA that regulates plant growth and development. Previous studies investigated that GA induces the transcription of genes that are involved in cell division and cell elongation occurring during plant growth [31]. GA also stimulates the induction of hydrolytic enzymes which are involved in the conversion of starch to sugar and the controlling of starch and sugar accumulation by GA can significantly influence plant growth and development [32]. Mostly the GA induces plant growth by alteration of certain genes expression. GA also plays a key role in several metabolic pathways that effecting plant growth such as chlorophyll biosynthesis, nitrogen metabolism and redistribution, and translocation of assimilates [28]. GA influences the differentiation of phloem fiber and enhances the length of bast fibers by inducing internode length. Reports show that a high level of GA results in primary phloem fiber elongation in the *Coleus blumei* plant and the length of differentiating internode is associated with the length of the primary phloem [33]. The increase in phloem fiber is associated with an increase in plant height and an increase in plant intermodal length [34]. Previous studies show that GA efficiently stimulates the lateral branches outgrowth in the *J. curcas* plant and it was investigated that increasing concentration increases shot branching [35]. In *Arabidopsis thaliana*, GA insensitive mutant shows a reduction in apical dominance and an increased number of axillary shoots [36]. GA sometimes shows an inhibitory role in lateral branching depending on plant species. In *Pisum sativum* plants, GA plays an inhibitory role in lateral bud formation [37]. The overexpression of the GA biosynthesis gene increased tillers or branches in *Paspalum distichum* and *Populus tremula*, suggesting that GA is significantly involved in the branching of these species [38,39,40]. However, GA-induced bud formation in citrus, sweet cherry, and rose [41,42,43]. Our results were consistent with the previously reported research; therefore, it is evident that GA is a positive regulator of stem elongation and buds and branches development. Our study further showed that GA-treated plants showed increased stem diameter, which suggested that GA is significantly involved in the secondary growth of *J. curcas*. Melanie Mauriat et al., (2011) reported that the expression of the GA biosynthesis gene in *Populus tremula* and *Nicotiana tabacum* enhanced the internode length and stem diameter [39]. Reports show that GA enhances the elongation and division of xylem and fiber cells in the vascular bundle region and increases cambium activity which increases stem diameter [44,45,46]. Compared to IAA and NPK, GA showed an increased number of leaves, branches, and stem diameter (Figure 2). Leaf number and growth are major determining factors contributing to shoot biomass and yield production. Leaves and branches number are regulated and controlled genetically and depend on developmental stage and species [47]. However, studies reported that growth regulators regulate leaves and branches development. In *Gladiolus grandifloras,* GA enhanced the leaves number and sprouting emergence compared with control plants [48]. Although there is a lack of data about GA induction of proximate composition in the *J. curcas* plant; however, it is reported that GA application increases carbohydrate contents in *Phalaenopsis* apex [49]. Reports show that GA accumulation enhances certain enzyme activity and increases cell wall plasticity which enhances membrane permeability and facilitates the uptake of mineral nutrients and transport of photosynthates [50,51,52,53]. The GA-facilitated uptake of nutrients and photosynthates transportation promotes plant growth and development. GA accelerates the cell cycle and starch hydrolysis to provide energy and a carbon skeleton for the synthesis of soluble sugar (carbohydrates) and other metabolites [54]. Our results showed that GA application increased protein contents in *J. curcas* which is in line with the results of Satendra Singh et al., (2014), they determined that GA_3_ significantly increased the total amino acid (protein contents) of *Phaseolus vulgaris* L. plant [55]. Taken together with these results, it is revealed that the exogenous application of GA enhances *J. curcas* growth and development mediated through the regulation of morphological, physiological, and biochemical parameters. 

*J. curcas* is a biodiesel-producing plant and possesses great characteristics of growing on barren land, low water, and harsh climatic condition. It is a nutrient-reactive plant and its requirement for nutrients varies with the soil fertility [56]. N, P, K, Zn, and B are the main fertilizers needed for the full-size growth of the plant to produce seeds [57,58]. N and P are the main nutrients that affect significantly the seed yield of *J. curcas* [59]. Researchers have suggested that, in the developmental stage of *J. curcas*, NPK is needed to build up the plant architecture such as root, stem, leaves, flowers, and seeds [57]. Analysis of our results revealed that NPK application to *J. curcas* showed significant differences in root shoot length, fresh dry weight, number of leaves, and branches. Nitrogen has a key role in chlorophyll biosynthesis and increases the rate of photosynthesis which results in increases in the dry biomass of the plant [60]. Similarly, phosphorus is involved in energy metabolism and photosynthesis while potassium plays a key role in carbohydrates and protein metabolism during plant growth [61,62]. In general, NPK fertilizer is easily absorbed by plants which play an important role in growth by supporting vegetative development including leaf, stem, and root development. Nitrogen is a building factor of protein, which greatly affect plant biomass [63]. Phosphorus is also used in protein and fat biosynthesis and transform adenosine diphosphate into adenosine triphosphate to generate energy [64]. Potassium enable CO2 during photosynthesis to enter through stomata, photosinthate transport, water, sugar, and protein and sugar synthesis [63]. Potassium availability increases energy that results into growth and development regulation [65]. Compared to GA and IAA, NPK 60 g showed the highest significant increase in total protein, carbohydrate, fat, moisture, and ash content percentage (Figure 3), which indicates that NPK higher concentration increases the proximate composition of *J. curcas* plant. Ali Sher Chandio et al., (2016) investigated that the highest amount of NPK application per hector showed maximum growth, fruit, and seed oil yield in the *J. curcas* plant [56]. Another research also revealed that the highest value of dry matter and seed fatty acid were obtained from NPK-treated *J. curcas* plants compared with non-treated plants [12]. Nitrogen has a key importance in plant growth than the other material as it plays a central role in many physiological and biochemical processes in plants [66]. It is a basic part of the structure of chlorophyll, protein, fats, and nucleic acid [66]. After nitrogen, phosphorus is an important macronutrient that affects plant growth. Without enough phosphorus, it is difficult for a plant to attain maximum growth and development as it has a key role in the storage and transfer of energy in plants [67,68]. Whereas potassium also plays a key role in metabolism and its application influence *J. curcas* seed oil and fatty acids [12]. All three macronutrients (NPK) are important growth inducers of plants but instead of using them separately, they are more efficient when used in combination [34]. Among the NPK, N plays a more important role in enhancing agriculture production by promoting chlorophyll, soluble protein, proline contents, and promoting fiber yield [69,70]. Compared to previous research, our study also determined that NPK application promoted *J. curcas* growth and development by enhancing morphological, physiological, and biochemical regime. 

## 4. Materials and Methods 

### 4.1. Experimental Design and Material Used

In the present study, six months old seedlings of equal size were collected from a common dealer in Multan, Pakistan. The seedling weight, stem width, and root length were different; therefore, the initial weight, stem width, and root length were measured and recorded. The whole experiment was conducted in pots and single seedlings were grown in each pot filled with an equal amount of soil and treated with hormones and fertilizer separately. The hormones Gibberellic acid GA3 (GA) and Indole acetic acid (IAA) were applied as 100 ppm and 250 ppm each to every single pot and NPK was applied 30 g and 60 g to each pot separately (mixed in the soil). Control plants were treated with only water. The plants were watered after two weeks with an equal amount of water. The data were collected after each week for four months and the whole experiment was conducted in triplicate. In fertilizer, nitrogen was 28%, phosphorus was 18% and potassium was 16%. 

### 4.2. Parameters Studied

In the present study, three main parameters were studied, i.e., physiological, morphological, and biochemical parameters. In physiological parameters, root, shoot length, and fresh and dry weight of root and stem were measured. In morphological parameters, the number of leaves, number of branches, and stem diameter were studied. In biochemical parameters, carbohydrates, proteins, lipids, moisture, and ash contents were studied. The data were taken every week until four months. 

### 4.3. Proximate Composition Analysis

To determine the effect of hormones and fertilizers on leaf moisture contents, the fresh leaves were randomly collected in triplicate from each treatment. The collected leaves were washed, weighed, and recorded as fresh weight (FW). The selected leaves were at 105 °C in the oven for 3 h. The fully dried leaves were again weighed and recorded as dried weight (Dw). The moisture content was calculated as the following:Moisture content (%) = (Fw − Dw) × 100 

To measure the ash content percentage, we collected 1 g of leaves and washed them to remove the contamination and the tissue dried. The leaves were burned in the muffle furnace at 450–550 °C to remove water and other volatile substances. The leaves samples were weighed before and after burning in the muffle furnace. The ash content was calculated as follows: Ash (%) = (M_ash_/M_dry_) × 100
where M_ash_ is the mass of fresh ash and M_dry_ is the mass of dry ash. 

To determine fat contents, we followed the method used by Aurea M. Almazan and Samuel O. Adeyeye (1998) [71]. About 1 g of fresh leaves were collected from *J. curcas* and dried in the oven at 60 °C for 15 h and then ground into fine powder. The powder was homogenized with hexane and extracted fat by using the Soxhlet method (AOAC, 1990). The extract was treated with methanolic sodium and methanolic boron trifluoride to convert the fatty acid into methyl ester, using the method followed by Paquot and Hautfenne in 1987 [72]. The fatty acid methyl ester in the hexane layer was dried at 90 °C by passing the nitrogen gas on its surface. The solution of hexane and methyl ester was filtered by a 45 µm filter and injected 1 µL into a Shimadzu gas chromatograph with a Perkin-Elmer PE-WAX capillary column (30 m × 025 mm) and flame ionization detector. The initial column temperature was 80 °C and increased to 260 °C kept for 7 min with the flow rate of gas (H_2_) being 30 cm/s. The percentage of fat was calculated based on the total area of fatty acids. All these determinations were carried out according to the Association of Official Analytical Chemists (AOAC, 1990). For crude protein determination, the Micro Kjeldahl method was followed according to the AOAC international [73]. About 1 g of the leaf samples was collected randomly then ground into a fine powder and digested with 15 mL H_2_SO_4_ by heating in the presence of K_2_SO_4_ and selenium using a heating block at 420 °C for 2 h. The digested sample was then neutralized by adding NaOH, to convert ammonium sulfate into ammonia, which is further distilled off and collected in a flask, and added boric acid was to form ammonium borate. The residual boric acid was further titrated with H_2_SO_4_ with the use of an endpoint indicator to determine the total nitrogen contents. The amount of total nitrogen in the raw material was multiplied by the traditional conversion factor of 6.25 and the specific conversion factor [74]. Carbohydrate content was determined by calculating the difference between the sum of all the proximate compositions from 100% [75].

### 4.4. Soil Analysis 

To find the nutrient deficiency in the soil, the soil was analyzed at the Barani Agriculture Research Center (BARS), Kohat. Two soil samples were randomly selected from the different sites of the field. The BARS report of soil is presented in Table 1. 

### 4.5. Statistical Analysis 

All experiments were performed in triplicate, and the data from each replicate were pooled. Data were analyzed using one-way ANOVA with Bonferroni post hoc tests (* shows *p* < 0.05, ** shows *p* < 0.01, and *** shows *p* < 0.001 significant difference). A completely randomized design was used to compare the mean values of different treatments. Data were graphically plotted, and statistical analyses were performed using the GraphPad Prism software (version 5.01, GraphPad, San Diego, CA, USA).

## 5. Conclusions

This study demonstrated that the exogenous application of phytohormones and fertilization of the *J. curcas* plant promoted growth and development. The results confirmed that different concentrations of GA, IAA, and NPK induced various parameters differentially. GA 250 ppm increased shoot root length, root fresh and dry weight, branches number, and stem diameter while GA 100 ppm increased leaves number as compared to IAA and NPK. While IAA 250 ppm increased stem diameter, and NPK increased proximate compositions compared to hormones. Our study concluded that optimum hormones and NPK level is essential for the efficient promotion of morphological, physiological, and biochemical aspects of the *J. curcas* plant. Furthermore, the hormones were used for scientific validation, however on commercial basis application of phytohormones are less economical than the fertilizers. Our current study opens an important research area for the future study to investigate, how to improve the indigenous phytohormones of *J. curcas* to improve its growth and development in the barren land. 

## Figures and Tables

**Figure 1 plants-11-03584-f001:**
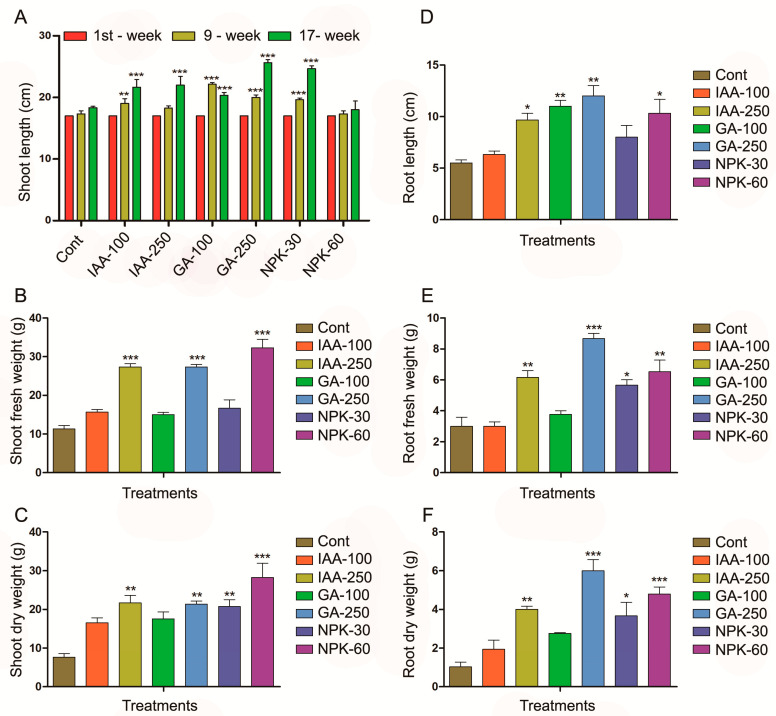
IAA, GA, and NPK regulate *J. curcas* plant soot, root length, shoot, root fresh and dry weight. (**A**) represents shoot length, (**B**) represent shoot fresh weight, (**C**) represent shoot dry weight. (**D**–**F**) represent foot length, root fresh weight, and root dry weight, respectively. Graphs show mean ± standard deviation, and asterisks show significant differences (* *p* ≤ 0.05, ** *p* ≤ 0.01, and *** *p* ≤ 0.001) according to two-way ANOVA and Bonferroni post hoc tests.

**Figure 2 plants-11-03584-f002:**
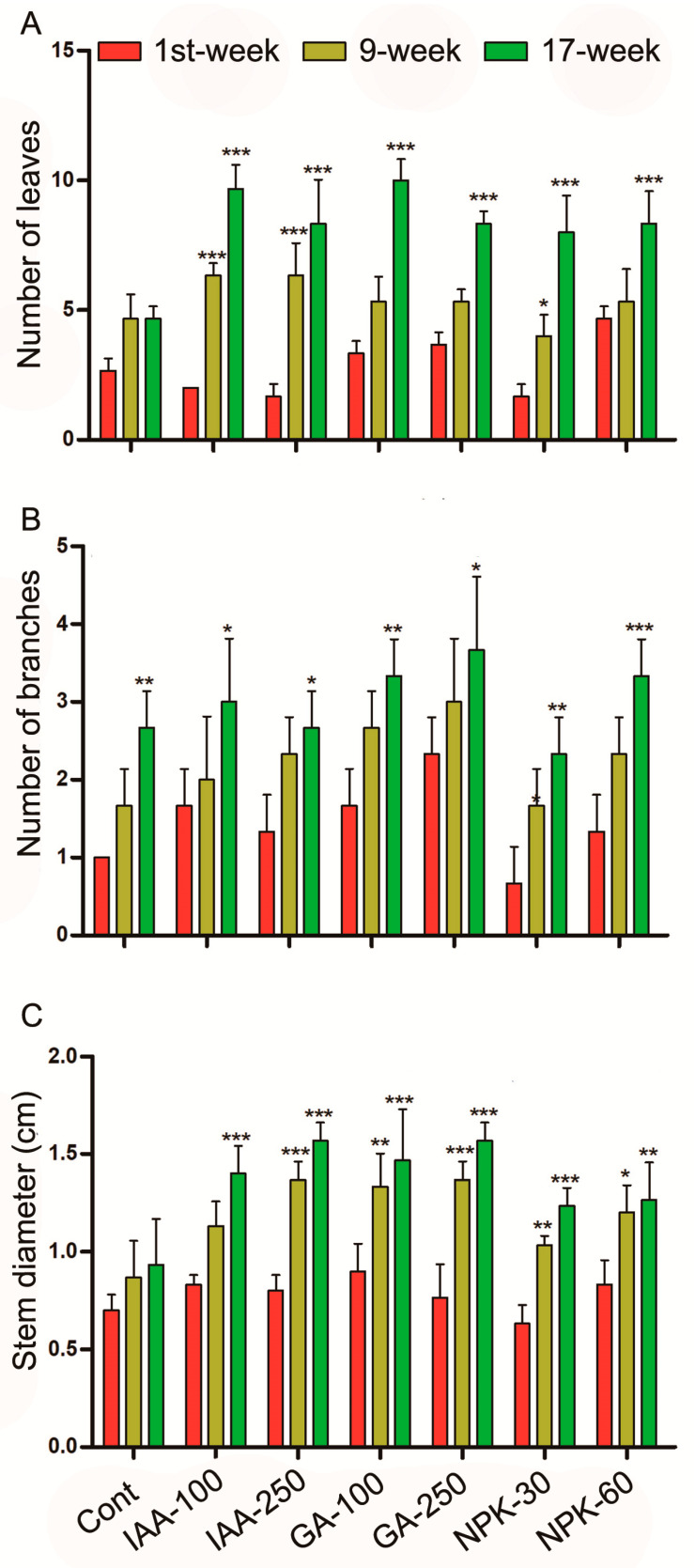
IAA, GA, and NPK regulate *J. curcas* leaves number, branches number, and stem diameter. (**A**) represents leaves number, (**B**) represents branches number, (**C**) represents stem diameter. Graphs show mean ± standard deviation, and asterisks show significant differences (* *p* ≤ 0.05, ** *p* ≤ 0.01, and *** *p* ≤ 0.001) according to two-way ANOVA and Bonferroni post hoc tests.

**Figure 3 plants-11-03584-f003:**
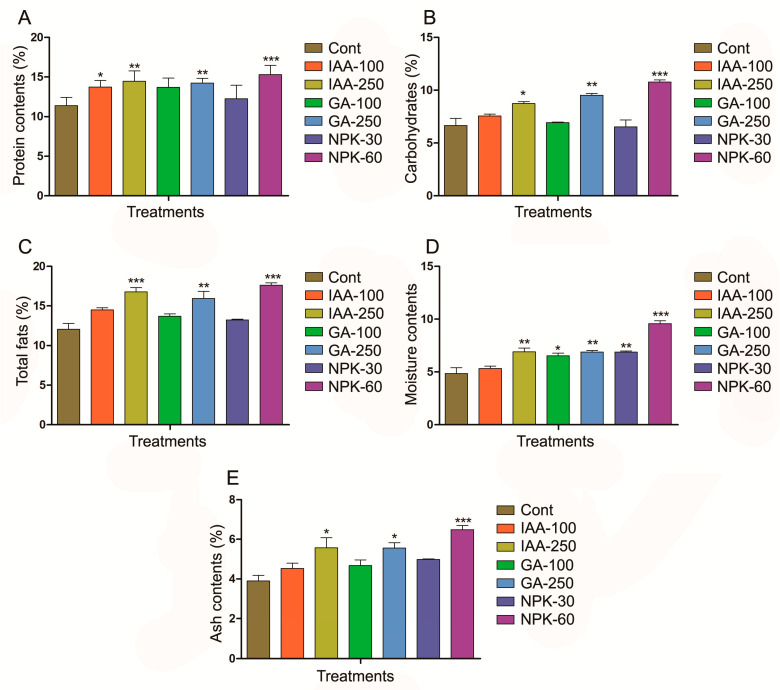
IAA, GA, and NPK induces biosynthesis of *J. curcas* plant proximate composition contents. (**A**) represents protein contents, (**B**) represents carbohydrate contents, (**C**) represents fat contents, (**D**) represents moisture contents, and (**E**) represents ash contents. Graphs show mean ± standard deviation, and asterisks show significant differences (* *p* ≤ 0.05, ** *p* ≤ 0.01, and *** *p* ≤ 0.001) according to two-way ANOVA and Bonferroni post hoc tests.

**Table 1 plants-11-03584-t001:** Parameters studied in soil analysis.

Sample No	pH	Electric Conductivity ds/m	CaCO_3_ %	Organic Manure %	Nitrogen %	Texture
1	7.51	0.55	14	0.69	0.0345	Clay loam
2	7.31	0.67	16	0.759	0.0379	Clay loam

## Data Availability

The data presented in this study are available on request from the corresponding author.

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
