# Peer review of "Exogenous Phytohormones and Fertilizers Enhance Jatropha curcas L. Growth through the Regulation of Physiological, Morphological, and Biochemical Parameters"

_plants, 2022, doi:10.3390/plants11243584_

Round 1
Reviewer 1 Report
Dear Authors,
The subject of this paper is important given that Jatropha is crop species with the ability to grow under drought conditions or poorly degraded soils. This kind of research has great potential for the further improvement of plant response against various stresses and subsequently crop production.
However, the presented Ms investigated the effect of exogenous phytohormones and fertilizers on growth parameters and proximate composition of Jatropha. Such results are quite obvious. Growth phytohormones, such as IAA, GA are known as substances stimulating shoot and root elongation, and in this way, they increase fresh and dry weight. So what about the novelty of the research?
The material and methods section is poorly written, e.g. the content of protein carbohydrate, fat, and ash was measured but I can’t find what tissues were used. The whole plant? Above ground part?
In my opinion, Ms in this form cannot be published.
Author Response
Reviewer #1
Dear Authors,
The subject of this paper is important given that Jatropha is crop species with the ability to grow under drought conditions or poorly degraded soils. This kind of research has great potential for the further improvement of plant response against various stresses and subsequently crop production.
However, the presented Ms investigated the effect of exogenous phytohormones and fertilizers on growth parameters and proximate composition of Jatropha. Such results are quite obvious. Growth phytohormones, such as IAA, GA are known as substances stimulating shoot and root elongation, and in this way, they increase fresh and dry weight. So what about the novelty of the research?
Reply: Thank you for your time and effort that you dedicated to providing your valuable feedback on our manuscript. We appreciate your insightful comments on our manuscript. We carried out the current study on the basis of our previous project failure. Previously, we have grown jatropha plants in a barren land where the soil nutrients was very low and the temperature was very high. Unfortunately, the whole plants were died after two months where the average temperature was 33-37°C. We design the current experiment to investigate that, whether the high temperature or the low nutrients effect the plant growth. Therefore, we conducted this experiment in the same location, same time and soil and applied growth promoting hormones and fertilizer and the plants were grown even at high temperature. We mentioned in our manuscript that jatropha can grow in nutrient deficient soil and high temperature but we predicted on the basis of our previous project that, Jatropha growth can be effected by exposure to low nutrients condition and high temperature, simultaneously. This experiment is of a great importance because on the basis of this experiment we can suggest that under the high temperature the growth regulator and the fertilizers can promote plant growth and development.
The material and methods section is poorly written, e.g. the content of protein carbohydrate, fat, and ash was measured but I can’t find what tissues were used. The whole plant? Above ground part?
Reply: Thank you for your important comment, we have revised the methodology and we hope that know it will be easily understandable for the readers.
Reviewer 2 Report
The manuscript “Exogenous phytohormones and fertilizers enhance Jatropha curcas L. growth through regulation of physiological, morphological and biochemical parameters” (plants-2026644) showed application IAA and GA in two concentration (100 and 250 ppm). In addition, two applications of NPK (30 and 60 g) were applied to Jatropha curcas, an important perennial plant. The author performed many analyses of the growth, morphological and physiological parameters.
The authors have done a large amount of work, employing various references and critical analysis based on a scientific method and structure. The introduction its ok, but M&M, the results and discussion topic is not good. For example, in some points phrases was confuse and English its not adequate. Major point its necessary corrections by adjusting in English syntax. In general, figures are not good qualities. For example, letter in y- and x-axis its different. In addition, I did not understand the purpose of applying hormones (IAA and GA) and then applying NPK. What is the interaction between them?
What is difference between GA, Gas GA3 or GA4? Please, standardization these terms, because, gibberellins is GAs (GA1 at GA136), but gibberellin (or gibberellic acid) is GA3, ok!
Please, use to template “Microsoft or Latex Template” of manuscript to figures, tables and references need adjust following “Author Instructions in Plants”.
-Figures, need correction following “Author Instructions” and displayed after first citation.
-Please, check all manuscript English language spelling. Phrases, syntax. Many phrases its confusion, in special results.
Point:
#1: There is a scope for improvement in the introduction section: a) additional emphasis on the significance of the study, b) scientific and economic contribution of the paper; c) prospectively to other plants to agronomic interest. Maybe a one paragraphs with potential economic by important this manuscript to science crop;
#2. Your hypothesis is not clear in last paragraph to introduction. Pease write “Our hypothesis was…”
#3: Please. All standardization of nomenclature equipment/reagents/software when necessary. Example: Fabricant, City, State, Country (three-letter). Check all manuscript.
#4 Please, check all scientific names and notations; In addition, I suggesting check English grammar by Native experts in all manuscript!!
#5. I’m suggesting to authors a Abbreviations list to clearance a many terms, that were not described in the manuscript.
#6. Please check for “Author Instruction” and standardization to manuscript, example, Figure/Figures, not Fig.
#8. Standardizing the units include space, i.e., 30 g or 30 g? 100 ppm or 100 ppm?;
#9. Please figures in text, not final o the manuscript; check for “Author Instruction”;
Specific points:
-Italic in scientific name; Please check in all manuscript;
L23. GA-GA3 or GA?
L31. GA 250 ppm?
L38. L39. Alphabetical order to keywords;
L42. Euphorbiaceae is not italic;
L80. “Which” not italic;
L218. Persea Americana (italic; its scientific name)
219. soybean,
L228-229. GA1 and GA3 subscripts;
245. Figure; Pease check all manuscript;
L254. Paly, not correct;
L263. Populus tremula (scientific name);
274-275. scientific name;
Figure 2. legend not bold;
Figure 3, what is fat?
Author Response
Reviewer # 2
Comments and Suggestions for Authors
The manuscript “Exogenous phytohormones and fertilizers enhance Jatropha curcas L. growth through regulation of physiological, morphological and biochemical parameters” (plants-2026644) showed application IAA and GA in two concentration (100 and 250 ppm). In addition, two applications of NPK (30 and 60 g) were applied to Jatropha curcas, an important perennial plant. The author performed many analyses of the growth, morphological and physiological parameters.
The authors have done a large amount of work, employing various references and critical analysis based on a scientific method and structure. The introduction its ok, but M&M, the results and discussion topic is not good. For example, in some points phrases was confuse and English its not adequate. Major point its necessary corrections by adjusting in English syntax.
Reply: We are thankful to you for the time you dedicated to provide us the feedback on our manuscript.
We revised the manuscript by the English native and we hope now it will be easily understandable to the readers.
In general, figures are not good qualities. For example, letter in y- and x-axis its different.
Reply: Thank you for your comment, we have revised the figures and the letters in Y and X-axis accordingly. The new figures are inserted into the manuscript.
In addition, I did not understand the purpose of applying hormones (IAA and GA) and then applying NPK. What is the interaction between them?
Reply: Thank you for your valuable comment. Our study is not related to the hormone and fertilizers interaction. Here we focused on the Jatropha growth pattern under exogenous hormones and fertilizers to find whether hormones or fertilizers increases the growth and development.
What is difference between GA, Gas GA3 or GA4? Please, standardization these terms, because, gibberellins is GAs (GA1 at GA136), but gibberellin (or gibberellic acid) is GA3, ok!
Reply: Thank you for your comment, in our study we used Gibberellic acid which is a GA3. We did not used gas GA3 and GA4 in our study. We mentioned in materials and methods that we used gibberellic acid GA3 and we mentioned with the name GA throughout the manuscript.
Please, use to template “Microsoft or Latex Template” of manuscript to figures, tables and references need adjust following “Author Instructions in Plants”.
Reply: Thank you for the comment, we revised figures and table and references according to the Journal format.
-Figures, need correction following “Author Instructions” and displayed after first citation.
-Please, check all manuscript English language spelling. Phrases, syntax. Many phrases its confusion, in special results.
Reply: We have revised the figures according to your above comment and inserted into the main text after the first citation. We have revised the manuscript for English correction with the English native.
Point:
#1: There is a scope for improvement in the introduction section: a) additional emphasis on the significance of the study, b) scientific and economic contribution of the paper; c) prospectively to other plants to agronomic interest. Maybe a one paragraphs with potential economic by important this manuscript to science crop;
Reply: thank you for your valuable suggestion, although we already included the significance and importance of the current study in the introduction however, we added some further informations suggested by the reviewer.
#2. Your hypothesis is not clear in last paragraph to introduction. Pease write “Our hypothesis was…”
Reply: Thank you for important comment, we revised the hypothesis.
#3: Please. All standardization of nomenclature equipment/reagents/software when necessary. Example: Fabricant, City, State, Country (three-letter). Check all manuscript.
Reply: Thank you for your important comment, we revised the whole manuscript according to the reviewer comment.
#4 Please, check all scientific names and notations; In addition, I suggesting check English grammar by Native experts in all manuscript!!
Reply: Thank you for your important comment, we carefully check the manuscript and revised the scientific names, and checked the grammatical errs by English native.
#5. I’m suggesting to authors a Abbreviations list to clearance a many terms, that were not described in the manuscript.
Reply: Thank you for your suggestion but there are very few terminologies which we already abbreviated where mentioned for the first time in the manuscript.
#6. Please check for “Author Instruction” and standardization to manuscript, example, Figure/Figures, not Fig.
Reply: Thank you, we revised the term according to the journal standard.
#8. Standardizing the units include space, i.e., 30 g or 30 g? 100 ppm or 100 ppm?;
Reply: We have revised the units according to the standard.
#9. Please figures in text, not final o the manuscript; check for “Author Instruction”;
Reply: We are apologetic, we did not get the point of reviewer, what the reviewer mean?
Specific points:
-Italic in scientific name; Please check in all manuscript;
Reply: thank you for your important comment, we have carefully screened the whole manuscript for scientific names and italicized it.
L23. GA-GA3 or GA?
Reply: thank you for you valuable comment, we used Gibberellic acid (GA3), we mention it with the name of GA in the manuscript.
L31. GA 250 ppm?
Reply: This is actually GA3- 250ppm but we mentioned as a GA 250 ppm.
L38. L39. Alphabetical order to keywords;
Reply: Revised.
L42. Euphorbiaceae is not italic;
Reply: Thank you, revised.
L80. “Which” not italic;
Reply: Revised.
L218. Persea Americana (italic; its scientific name)
Reply: Revised.
- soybean,
Reply: Revised.
L228-229. GA1 and GA3 subscripts;
Reply: Revised.
- Figure; Pease check all manuscript;
Reply: thank you, convert Fig to Figure.
L254. Paly, not correct;
Reply: thank you, revised.
L263. Populus tremula (scientific name);
Reply: Revised.
274-275. scientific name;
Reply: Revised.
Figure 2. legend not bold;
Reply: Thank you, revised figure legends added to the end of manuscript after references.
Figure 3, what is fat?
Reply: Fat is the total fatty acid contents, we revised in figure 3 and all the figures are added into the manuscript.
Reviewer 3 Report
Respected Sir,
Thanks for considering me to review the manuscript titled “Exogenous phytohormones and fertilizers enhance Jatropha curcas L. growth through regulation of physiological, morphological and biochemical parameters”
The present study was conducted to improve the growth and development of J. curcas plant by exogenous application of gibberellin (GA), indole acetic acid (IAA), and fertilizer (nitrogen, phosphorus, potassium).
As the study has some novelty, so it should be improved and published. The manuscript needs some minor revisions addressed in the following comments. Overall, the MS must be revised to avoid the typos mistakes. All the scientific names of plants and microbes must be in italic.
Abstract: does not cover all the study sides. Add more details to the study. The aim of the study is shown in a very lake presentation
Introduction covered the research point from all sides. But several references are old and need to be updated
The objective of the study needs to be rewritten to show the novelty of the study.
Methods: The experiment and the chemical analysis were conducted correctly, however, they lake the details to repeat them. The description of analysis methods is written in a short way and they need to more descriptive to enable other to repeat the experiment.
Results: The description of results is satisfied.
Discussion: I think you should focus on the mechanism of phytohormones and fertilizers in increasing the growth.
Author Response
Reviewer #3
Respected Sir,
Thanks for considering me to review the manuscript titled “Exogenous phytohormones and fertilizers enhance Jatropha curcas L. growth through regulation of physiological, morphological and biochemical parameters”
The present study was conducted to improve the growth and development of J. curcas plant by exogenous application of gibberellin (GA), indole acetic acid (IAA), and fertilizer (nitrogen, phosphorus, potassium).
As the study has some novelty, so it should be improved and published. The manuscript needs some minor revisions addressed in the following comments. Overall, the MS must be revised to avoid the typos mistakes. All the scientific names of plants and microbes must be in italic.
Reply: Thank you for your valuable comment, the whole manuscript were revised for typos and all the scientific names were italicized.
Abstract: does not cover all the study sides. Add more details to the study. The aim of the study is shown in a very lake presentation
Reply: Thank you for your important comment, we included the aim of our study in the last paragraph of the introduction. The reviewer suggested adding more detail of the study but unfortunately, we did not clearly understand what kind of detail needed to be added. However, we have already included all the results in the abstract and if we add more introduction of the study then the abstract will be too much descriptive. However, we will appreciate the reviewer if he/she suggest us which section is needed to be add more information.
Introduction covered the research point from all sides. But several references are old and need to be updated
Reply: We updated the old references, thank you.
The objective of the study needs to be rewritten to show the novelty of the study.
Reply: Thank you for your important suggestion, we revised the objective of our study in the last paragraph of the introduction.
Methods: The experiment and the chemical analysis were conducted correctly, however, they lake the details to repeat them. The description of analysis methods is written in a short way and they need to more descriptive to enable other to repeat the experiment.
Reply: Thank you for your important comment, we revised the methodology according the reviewer comment.
Results: The description of results is satisfied.
Reply: Thank you for your comment.
Discussion: I think you should focus on the mechanism of phytohormones and fertilizers in increasing the growth.
Reply: Thank you for your valuable comment, we have revised the discussion part and added some more information about the mechanism of hormones and fertilizers in plant growth.
Round 2
Reviewer 2 Report
I consider the authors made important changes in the manuscript and it was highly improved. However, minor point was necessary correction. GA3(subscript)
L673. Glycine max. Check all, scientific names.
L704. CO2(subscript)
References: L270;
Author Response
I consider the authors made important changes in the manuscript and it was highly improved. However, minor point was necessary correction. GA3(subscript)
Response: Thank you for your comment, GA3 subscripted.
L673. Glycine max. Check all, scientific names.
Response: Revised.
L704. CO2(subscript).
Response: Revised.
References: L270;
Response: Thank you for your comment, however, we checked L270 and we could not notice any changes, we think there is some misunderstanding in line number.
Reviewer 3 Report
The authors corrected the MS and it can be accepted
Author Response
The authors corrected the MS and it can be accepted.
Response: Thank you for your time dedicated to reviewing our manuscript and acceptance for publication.